# A Large-Scale 3D Study on Transport of Humic Acid-Coated Goethite Nanoparticles for Aquifer Remediation

**Milica Velimirovic** [1,2], **Carlo Bianco** [3], **Natalia Ferrantello** [3], **Tiziana Tosco** [3], **Alessandro Casasso** [3], **Rajandrea Sethi** [3,*], **Doris Schmid** [1], **Stephan Wagner** [1,4], **Kumiko Miyajima** [5], **Norbert Klaas** [5], **Rainer U. Meckenstock** [6], **Frank von der Kammer** [1] and **Thilo Hofmann** [1,*]

[1] Department of Environmental Geosciences, Centre for Microbiology and Environmental Systems Science, University of Vienna, Althanstrasse 14, 1090 Vienna, Austria; milica.velimirovicfanfani@ugent.be (M.V.); doris.schmid@a1.net (D.S.); stephan.wagner@ufz.de (S.W.); frank.kammer@univie.ac.at (F.v.d.K.)

[2] Department of Chemistry, Atomic & Mass Spectrometry–A&MS Research Group, Campus Sterre, Ghent University, Krijgslaan 281-S12, 9000 Ghent, Belgium

[3] Department of Environmental, Land and Infrastructure Engineering (DIATI), Politecnico di Torino, corso Duca degli Abruzzi, 24-10129 Torino, Italy; carlo.bianco@polito.it (C.B.); natalia.ferrantello@polito.it (N.F.); tiziana.tosco@polito.it (T.T.); alessandro.casasso@polito.it (A.C.)

[4] Department of Analytical Chemistry, UFZ-Helmholtz Centre for Environmental Research, Permoserstrasse 15, 04318 Leipzig, Germany

[5] VEGAS—Research Facility for Subsurface Remediation, University of Stuttgart, Pfaffenwaldring 61, 70569 Stuttgart, Germany; kumiko.miyajima@googlemail.com (K.M.); norbert.klaas@iws.uni-stuttgart.de (N.K.)

[6] Environmental Microbiology and Biotechnology, University Duisburg-Essen, 45141 Essen, Germany; rainer.meckenstock@uni-due.de

\* Correspondence: rajandrea.sethi@polito.it (R.S.); thilo.hofmann@univie.ac.at (T.H.); Tel.: +39-011-090-7735 (R.S.); +43-1-4277-53320 (T.H.)

**Abstract:** Humic acid-coated goethite nanoparticles (HA-GoeNPs) have been recently proposed as an effective reagent for the in situ nanoremediation of contaminated aquifers. However, the effective dosage of these particles has been studied only at laboratory scale to date. This study investigates the possibility of using HA-GoeNPs in remediation of real field sites by mimicking the injection and transport of HA-GoeNPs under realistic conditions. To this purpose, a three-dimensional (3D) transport experiment was conducted in a large-scale container representing a heterogeneous unconfined aquifer. Monitoring data, including particle size distribution, total iron (Fe$_{tot}$) content and turbidity measurements, revealed a good subsurface mobility of the HA-GoeNP suspension, especially within the higher permeability zones. A radius of influence of 2 m was achieved, proving that HA-GoeNPs delivery is feasible for aquifer restoration. A flow and transport model of the container was built using the numerical code Micro and Nanoparticle transport Model in 3D geometries (MNM3D) to predict the particle behavior during the experiment. The agreement between modeling and experimental results validated the capability of the model to reproduce the HA-GoeNP transport in a 3D heterogeneous aquifer. Such result confirms MNM3D as a valuable tool to support the design of field-scale applications of goethite-based nanoremediation.

**Keywords:** large-scale 3D study; humic acid-coated goethite nanoparticles; transport modelling; MNM3D

## 1. Introduction

In recent years, there has been considerable interest in the use of nanomaterials for in situ remediation of contaminated groundwater as a non-invasive, flexible, and cost-effective technology [1–3]. The application of nanomaterials for in situ degradation, transformation, or immobilization of pollutants consists in the generation of a reactive zone by injecting a suspension of engineered nanoparticles in the vicinity of the contamination source [4]. The most promising and commonly studied engineered nanoparticles include nanoscale zerovalent iron (nZVI) [5–8], carbon supported iron [9], and iron oxide (FeOx) (e.g., goethite (a-FeOOH)) [10].

A promising approach for aquifer restoration is represented by the subsurface injection of goethite nanoparticles (GoeNPs) to stimulate intrinsic microbial iron reduction and consequently enhance the microbial oxidation of numerous recalcitrant groundwater contaminants (such as BTEX compounds) [10]. Besides stimulating the biodegradation of organic groundwater contaminants, GoeNPs are also capable of adsorbing heavy metals (e.g., As(V)) in oxic groundwater systems [11,12]. The emplacement of GoeNPs requires the production of a suspension that is stable against aggregation and characterized by a low deposition (or filtration) rate to prevent the clogging and fracturing of the porous medium [8,13,14]. To reduce the aggregation of GoeNPs and maximize their mobility in the subsurface after injection, GoeNPs are usually coated by humic acids (HA-GoeNPs) [10,15].

Previous work demonstrated the effective transport of HA-GoeNPs in a 1D column [16] and the possibility to use 1D and 2D models to support the interpretation of the laboratory experiment and the design of pilot interventions [14,17]. Nevertheless, despite laboratory experiments being useful for the comprehension and characterization of the physico-chemical mechanisms governing particle transport in porous media, they cannot be fully representative of the complex conditions observed during field-scale applications (e.g., porous medium heterogeneity). Controlled large-scale 3D experiments, and consequent model interpretation of data, are therefore required for a reliable upscaling of laboratory evidences to conditions more representative of real aquifers. However, no such 3D experiments have been reported in the literature for HA-GoeNP injection and transport up to now.

Here, we report a field-like 3D laboratory experiment, where a suspension of HA-GoeNPs was injected in a large-scale container ($9.0 \times 6.0 \times 4.5$ m; L $\times$ W $\times$ H) reassembling a heterogeneous unconfined aquifer. The experiment was performed under well-controlled and reproducible conditions and a high-resolution monitoring system was installed to get a reliable spatial distribution of HA-GoeNPs during and after the injection. This was the first time that goethite particle injection and transport have been tested in such a realistic setup and controlled environment, demonstrating that HA-GoeNP delivery is feasible for aquifer restoration. For a quantitative analysis of the results, a flow and transport model of the container was built using the numerical code Micro and Nanoparticle transport Model in 3D geometries (MNM3D) [17]. The kinetic parameters implemented in the model to describe goethite transport in porous media were derived from the interpretation of simple column tests (reported in Supporting Information) and then applied in forward mode to predict the particle behavior within the container. Modeling and experimental results were finally compared to validate, for the first time, the capability of the model to reproduce the HA-GoeNP transport in a 3D heterogeneous aquifer.

## 2. Materials and Methods

### 2.1. Humic Acid-Coated Goethite Nanoparticles and Particle Suspension

The suspension of HA-GoeNPs was prepared according to US patent 8921091B2 [15] and provided by the University of Duisburg-Essen (Germany). The suspension of HA-GoeNPs consisted of 10 g iron (Fe) L$^{-1}$ and 12 g total organic carbon (TOC) L$^{-1}$. The hydrodynamic radius and the zeta potential of HA-GoeNPs were determined in a (1:10 diluted) suspension by Zetasizer Nano ZS (Malvern instruments Ltd., Worcestershire, UK) The $\zeta$ potential of HA-GoeNPs was strongly negative ($-56 \pm 2$ mV at pH = 9.2, calculated from the electrophoretic mobility by using the Smoluchowski relationship) revealing the role of humic acid in particle stabilization and its importance for in situ

applications. The suspension of HA-GoeNPs is polydisperse and it is characterized by a broad particle size distribution, as previously observed by Montalvo et al. [10], for particles produced following the same approach. The intensity-based (Z-average) mean diameter of 197 ± 2 nm was measured using the Zetasizer Nano ZS. The HA-GoeNPs size was also measured with an asymmetrical flow field-flow fractionation (AF4) combined with a multi-angle light scattering (MALS) (Wyatt Technology, Dernbach, Germany) and an inductively coupled plasma-mass spectrometer (ICP-MS) (Agilent 8800, Agilent, Santa Clara, CA, USA) to identify the particle composition (specifications of the AF4 and the ICP-MS are summarized in Table S1). A hydrodynamic diameter of 164 nm was measured, thus confirming DLS results (for more details, see Figure S1). Sedimentation tests to assess the suspension stability of HA-GoeNPs were performed by monitoring the transmission of monochromatic light through the suspension at a wavelength of 880 nm (TurbiScan LAB, Quantachrome, Odelzhausen, Germany). The results confirmed the good stability of the HA-GoeNP suspension, most likely due to their small size and electrostatic repulsion provided by the modification with humic acids. As a consequence, the particles were stable for the whole duration of the transport experiment (>24 h).

## 2.2. The Large-Scale Three-Dimensional (3D) Container

The transport experiment was performed under controlled conditions in a large-scale three-dimensional (3D) container (Figure 1) installed at the VEGAS research facility, University of Stuttgart (DE). The container is made of high-grade stainless steel, has a rectangular section and a total volume of 243 m$^3$ (9.0 × 6.0 × 4.5 m; L × W × H). The container was filled with medium and coarse sand with a block structure: 60 sand blocks randomly distributed in three layers (from top to bottom: layer 1, layer 2, and layer 3), respectively 1.75, 1.45, and 1.3 m high. The hydrodynamic properties of the porous media are reported in Table 1. In particular, the hydraulic conductivity values of $4 \times 10^{-4}$ m/s and $3.5 \times 10^{-3}$ m/s, respectively, for medium and coarse sand, were obtained in previous studies from constant head permeameter tests; porosity and dispersivity values were calibrated against a tracer test (data not reported; an example of tracer test results is reported in Figure S2).

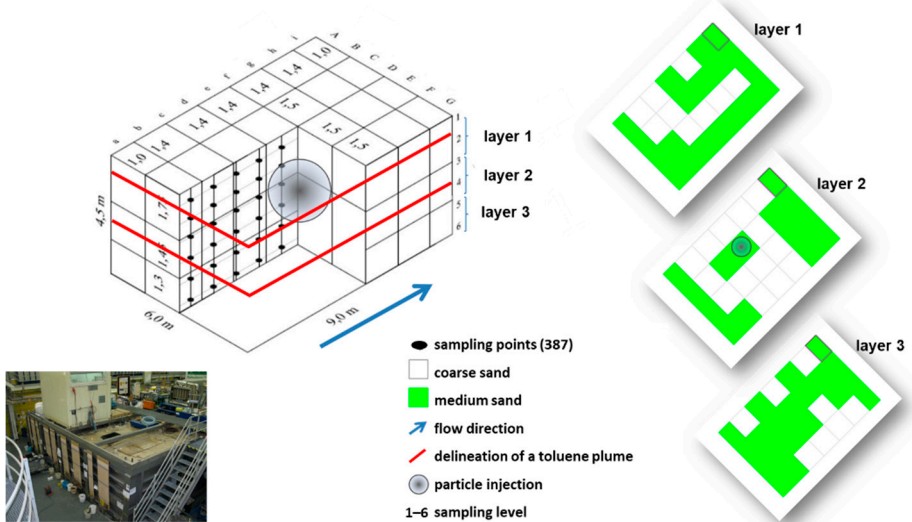

**Figure 1.** Container layout (plan and cross-section view) indicating injection well and sampling ports.

The liquid sampling ports are located on the left-hand side wall of the container (with respect to the groundwater flow). A total of 378 sampling points are located in six sampling planes (from top to bottom: level 1–6), nine sampling rows (a–i, along the flow direction), and seven sampling columns (A–G, perpendicularly to flow direction, from left to right). Two additional sampling points are included in the inflow and outflow reservoirs. The background flow, simulating natural groundwater flow, was established in the container by injecting a constant discharge rate, Q, through 12 equally

spaced, fully screened wells, located close to the upstream wall of the container (approximately 0.3 m far from the wall), and extracting the same discharge rate from a draining reservoir at the downstream boundary. For this test, a constant discharge rate, Q, of 3.1 m³/d was injected for the entire duration of the test. The discharge rate was equally distributed through the 12 wells (i.e., approximately 0.25 m³/d each well), resulting in a Darcy flux q of approximately 0.14 m/d. The low discharge rate applied at the injection wells, associated with a sufficient wall–well distance, ensured the flow field in the central portion of the container, where particle injection was performed, to be unaffected by any disturbance in the individual wells (as demonstrated by the simulated undisturbed flow field reported in Figure S3). At the downstream reservoir, a constant head of 3.7 m was reached at steady state conditions. The average depth of the water table was 0.8 m below ground level (bgl).

**Table 1.** Characteristics of the heterogeneous aquifer (medium and coarse sand).

|  | Medium Sand | Coarse Sand |
| --- | --- | --- |
| Grain size distribution (m) | $0$–$4 \times 10^{-3}$ | $0.2$–$8 \times 10^{-3}$ |
| Sand particle density (kg/m³) | $2.65 \times 10^{3}$ | $2.65 \times 10^{3}$ |
| Bulk density (kg/m³) | $1.72 \times 10^{3}$ | $1.59 \times 10^{3}$ |
| Hydraulic conductivity (m/s) | $4 \times 10^{-4}$ | $3.5 \times 10^{-3}$ |
| Porosity | 0.35 | 0.4 |
| Longitudinal dispersivity (m) | 0.02 | 0.04 |

*2.3. Transport Experiment*

2.3.1. Injection of HA-GoeNPs

The suspension of HA-GoeNPs ($c_{particle}$ = 10 g/L) was injected into the 3D container through one well (ID = 3″) located within layer 1 and 2 (L screen = 2.2 m, x = 3.48 m and y = 3.90 m). The impeller (Nirostar 2000-D–400 V, Sinntec Schmiersysteme GmbH, DE) pump injected a volume $V_{HA\text{-}GoeNPs}$ = 6 m³ with a flow of $Q_{inj}$ = 0.7 m³/h, corresponding to an injection time of approximately 8 h (see Figure S4). During injection, the inlet tank was continuously mixed to guarantee a uniform suspension.

2.3.2. Monitoring Set Up

In order to monitor the transport of HA-GoeNPs, 20 mL water samples were collected in levels 2, 3 and 4 from selected sampling ports every 30 min during the injection (approximately 8 h), as well as at 12, 24, and 48 h from the beginning of the test. The location of the sampled ports (46 in total) was selected based on the expected particle migration path and is identified in Figure S5. HA-GoeNP concentrations were indirectly determined via turbidity measurements performed with the turbidimeter 2100N IS (Hach Lange, Germany), following the ISO7027 Method. The linear correlation between concentration and turbidity was verified with a five-point calibration in the range from 0 (Milli-Q water) to 1000 NTU (Figure S6).

The particle size fractionation and the subsequent particle sizing were performed using an Eclipse Dualtec asymmetrical flow field-flow fractionation (AF4) system (Wyatt Technology, Dernbach, Germany). The carrier solution was delivered by an Agilent1200 series quaternary HPLC pump equipped with a micro vacuum degasser. The outflow was analyzed using a multi-angle light scattering (MALS) detector (DAWN® EOSTM, Wyatt Technology Europe GmbH, Dernbach, Germany) and an ICP-MS (Agilent 8800, Agilent, Santa Clara, CA, USA).

The initial suspension of HA-GoeNPs used in this study was further diluted with Milli-Q water (Millipore, Elix®5-Milli-Q® Gradient A10, Darmstadt, Germany) to the desired working concentration to perform the measurements.

### 2.4. Modeling Approach for Assessing the Transport of HA-GoeNPs

The transport of colloidal particles in saturated porous media is usually modeled by a modified advection-dispersion equation, which describes the non-equilibrium interactions between the particles suspended within the carrier fluid (mobile phase) and the grains of the porous medium (solid phase) [18–20]. For a generic domain, the following system of equations can be considered

$$\frac{\partial}{\partial t}(\varepsilon C) + \sum_N \frac{\partial}{\partial t}(\rho_b S_N) + \frac{\partial}{\partial x_i}(q_i C) - \frac{\partial}{\partial x_i}\left(\varepsilon D_{ij}\frac{\partial C}{\partial x_j}\right) - q_s C_s = 0$$
$$\frac{\partial}{\partial t}(\rho_b S_N) = \varepsilon k_{a,N}\left(1 + A_N S_N^{B_N}\right)C - \rho_b k_{d,N} S_N \tag{1}$$

where $\varepsilon$ is the porosity of the aquifer (-), $q_i$ is the Darcyan flow velocity along the $i^{\text{th}}$ direction (LT$^{-1}$), $C$ is the colloid concentration in the liquid phase (ML$^{-3}$), $S_N$ is the colloid concentration retained on the solid phase on the $N$-th site (MM$^{-1}$), $D_{ij}$ is the dispersion coefficient tensor (L$^2$T$^{-1}$), $\rho_b$ is the bulk density of the solid matrix (ML$^{-3}$), $q_s C_s$ is the mass flux of the source and sink terms (per unit volume of the aquifer) (MT$^{-1}$), $k_{a,N}$ and $k_{d,N}$ are, respectively, the attachment and detachment coefficients for the $N$-th site (T$^{-1}$). $A_N$ and $B_N$ are semi-empirical coefficients (-) of a generic formulation proposed by Tosco and Sethi [21] to model different linear and non-linear deposition mechanisms depending on the type of physical–chemical interactions considered (i.e., linear deposition, blocking, ripening).

The first equation describes the transport of suspended particles in the mobile phase, due to advection, hydrodynamic dispersion and particle–soil interaction mechanisms (due to a mass exchange between the liquid and solid phase) that result into particle filtration, deposition on and release from the porous media. The second equation represents the mass exchange term that takes into account particle–soil interactions, which are strongly dependent on pore water geochemical and hydrodynamic parameters (e.g., ionic strength, flow velocity) [22–25].

In this work, the HA-GoeNP injection and transport within the large-scale container was simulated using MNMs 2018 and MNM3D, which both provide a numerical solution to the system of Equation (1), respectively, in 1D and 3D domains. The Micro- and Nanoparticle transport, filtration and clogging Model Suite (MNMs 2018) (www.polito.it/groundwater/software/mnms) is a finite-difference code for the simulation of solute and colloidal transport in 1D saturated porous media [8,26,27]. MNMs 2018 was here applied for the interpretation of column transport tests of nanoparticle transport at laboratory scale. Micro and Nanoparticle transport Model in 3D geometries (MNM3D) is a numerical tool developed by Bianco, Tosco and Sethi [17] to simulate micro- and nanoparticle injection and transport in complex and three-dimensional domains. MNM3D is based on RT3D [28], and was built and run in a customized release of Visual Modflow.

An integrated experimental-modeling approach was applied to infer the transport dynamics governing the HA-GoeNP migration into the container, starting from experimental results of laboratory column transport tests, and up-scaling the results by means of MNMs 2018 and MNM3D. The strategy was developed through the following main steps [17]:

1.  Column transport tests were run in the laboratory by injecting the HA-GoeNPs in saturated columns packed with a sand similar to the one present in the large tank container;
2.  Particle transport kinetic parameters (attachment/detachment coefficients, etc.) were obtained by fitting the results of the laboratory tests using MNMs 2018 (the procedure is further described in Section 2.4.1);
3.  A numerical model of the large-scale container was built in Visual Modflow.
4.  A tracer test was performed in the large-scale container (Figure S2). The experimental results were used to calibrate porosity and longitudinal dispersivity (Table 1); a ratio of longitudinal to transveral dispersivity equal to 0.1 was assumed;
5.  The HA-GoeNPs were injected into the large-scale container;
6.  The HA-GoeNP injection was simulated by using MNM3D, the particle transport parameters obtained from column tests, and numerical results compared with experimental ones.

The strategy for the derivation of the transport coefficients and the simulation of the iron-oxide injection into the container are further discussed in Sections 2.4.1 and 2.4.2.

### 2.4.1. Derivation of the Kinetic Parameters for the Simulation of HA-GoeNPs Transport in A Large-Scale 3D Container

In order to have a detailed physical–chemical characterization of the HA-GoeNP slurry, an experimental-modeling strategy was followed:

- Two column tests were performed at two different Darcy velocities ($v_1$ = 100 m/d, $v_2$ = 10 m/d). Columns, having a diameter of 0.025 m and a length of 0.22 m, were wet-packed with the same sand used for the preparation of the container to mimic case-specific conditions. Nanoparticles were injected at a concentration $C_0$ of 10 g/L;

- The physical–chemical nature of the particle–soil interactions was determined by a qualitative analysis of the breakthrough curves shape, which suggested the mechanical filtration to be the dominant process. As a consequence, the particle transport in the column was modeled using Equation (1) with h = 1 (one-dimensional domain), and the following formulation was adopted for the interaction with the solid phase (i.e., second equation in Equation (1)), corresponding to an irreversible linear deposition ($A = 0$, $B = 1$, $k_d = 0$)

$$\frac{\partial}{\partial t}(\rho_b S) = \varepsilon k_a C \tag{2}$$

- A least-squares fitting of the column test results was conducted (Figure S7 in Supporting Information) to obtain the case-specific kinetic parameters. Modeled and experimental breakthrough curves are reported in the Supporting Information of Figure S8, where details of the numerical solution of Equations (1) and (2) are also provided. Two values of $k_a$ were found ($k_{a,high}$ = 3.75 × 10$^{-5}$ 1/s, $k_{a,low}$ = 2.63 × 10$^{-5}$ 1/s), respectively, for the high- and low-velocity test. An average value of $k_a$ equal to 3.19 × 10$^{-5}$ 1/s was then considered and implemented in MNM3D;

- Particle transport in the large-scale container was simulated using MNM3D to numerically solve Equation (1) with a linear irreversible deposition kinetics (i.e., Equation (2) replaces the second term of the system of Equation (1)). The attachment coefficient in the 3D model was imposed equal to the value derived from column test modeling, i.e., $k_a$ = 3.19 × 10$^{-5}$ 1/s. A third type boundary condition was applied at the particle injection well, and a second type zero-gradient boundary condition was applied at the outlet of the domain.

### 2.4.2. The Simulation of Water Flow and HA-GoeNPs Transport in Large-Scale 3D Container

The model of the large-scale container is a parallelepiped of size 9.0 × 6.0 × 4.5 m (Figure 2). Container was divided into homogeneous sub-domains characterized by different values of hydraulic conductivity, in accordance to the block structure of the physical tank presented in Section 2.2.

The domain was discretized with a regular grid of 900 × 60 × 20 cells (L × W × H), coherently with model size. The discretization was finer in the proximity of the NP injection well. A total of 378 monitoring points were placed in six sampling planes (1–6), nine sampling rows (a–i) and seven sampling columns (A–G), reproducing the location of the sampling ports shown in Figure 1 (e.g., the monitoring point 2dC is the sampling port on the 2nd level of the piezometer dC). The boundary conditions reproduced the configuration of the experimental container, as described in Section 2.2. The background groundwater flow was obtained by implementing 12 fully penetrating wells positioned upstream, each one injecting a constant flow rate of 0.25 m$^3$/d, and by imposing a constant hydraulic head of 3.7 m downstream (1st kind Dirichlet flow boundary condition) (Figure 2b). No-flow boundaries were imposed on all other domain walls (including the upstream wall). The simulated background flow field is reported in Figure S3. The steady state flow generated by the wells and the constant head was used as an initial condition of the particle injection simulation, solved by MNM3D. To this aim,

a constant discharge rate of 0.7 m$^3$/h with a HA-GoeNP concentration $C_0$ = 10 mg/L was imposed for 8.5 h through a well that was positioned at x = 3.48 m and y = 3.90 m and screened at depth z = 1.7–3.7 m. The final step of the simulation included the restoration of background flow with no further particle injection. The simulation setup is summarized in Table 2.

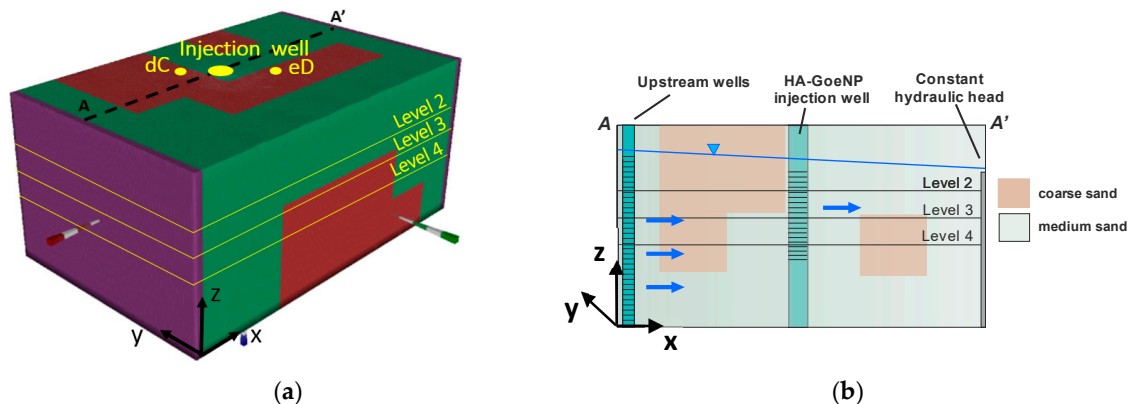

(**a**)          (**b**)

**Figure 2.** Model of aquifer realized in Micro and Nanoparticle transport Model in 3D geometrics (MNM3D) (**a**), model conceptualization along the transversal section A-A' (**b**).

**Table 2.** Flow and transport simulation setup.

|  | **Flow Simulation** | **Transport Simulation** |
|---|---|---|
| **Solver type** | MODFLOW 2005 (WHS method) | MNM3D (GCG method) |
| **Numerical discretization** | Central finite difference | Central finite difference |
| **State** | Steady | Transient |
| **Time-step** | - | Automatic control (Multiplayer 1.02) |
| **Error tolerance** | Residual criterion ($10^{-5}$) | Relative convergence criterion ($10^{-4}$) |

Modeled and experimental results were compared in terms of breakthrough curves detected at the monitoring points located in the container.

## 3. Results and Discussion

Experimental and simulated concentrations were compared in correspondence of the selected 46 monitoring points homogeneously spread within the container (Figure 3). Figure 4 illustrates experimental (red points) and modelled (black lines) breakthrough curves at two observation points 2dC and 3eD, situated respectively in level 2 and 3 at the coordinates x2dC = 3.5 m, y2dC = 3.84 m, z2dC = 3.31 m and x3eD = 4.5 m, y3eD = 2.9 m, z3eD = 2.62 m.

### 3.1. Transport of HA-GoeNPs in A Heterogenous Unconfined Aquifer–Experimental Concentration

Monitoring data including Fe$_{tot}$ content revealed the transport of particles toward the high permeability zone at level 2–4 (Figure 3), showing that the permeability of porous media has a dominating impact on the particle transport after injection. The breakthrough of the HA-GoeNPs showed a reasonable radius of influence of approximately 2 m. This is in accordance with the previously published lab-scale data by Tosco, Bosch, Meckenstock and Sethi [16] where HA-GoeNPs showed a high mobility. The shape of the breakthrough curves based on the measured Fe$_{tot}$ data correspond well with the time of the injection. Moreover, the breakthrough of the HA-GoeNPs of up to 80%–90% during the injection (monitoring point 2dC and 3eD, Figure 4) reveals the possible sedimentation or attachment of HA-GoeNPs to the sand material. A decline in concentration at both monitoring points is observed after injection, when only the background flow is re-stablished, indicating that particles are progressively filtered in the porous medium, tend to sediment over time, and as a consequence the mobility progressively declines. Similar trends are observed at all monitoring points (see other

examples in Supporting Information, in Figure S8). An unexpectedly high concentration was registered at 48 h at the monitoring point 3eD, after the decline observed at previous times, which could be considered an outlier.

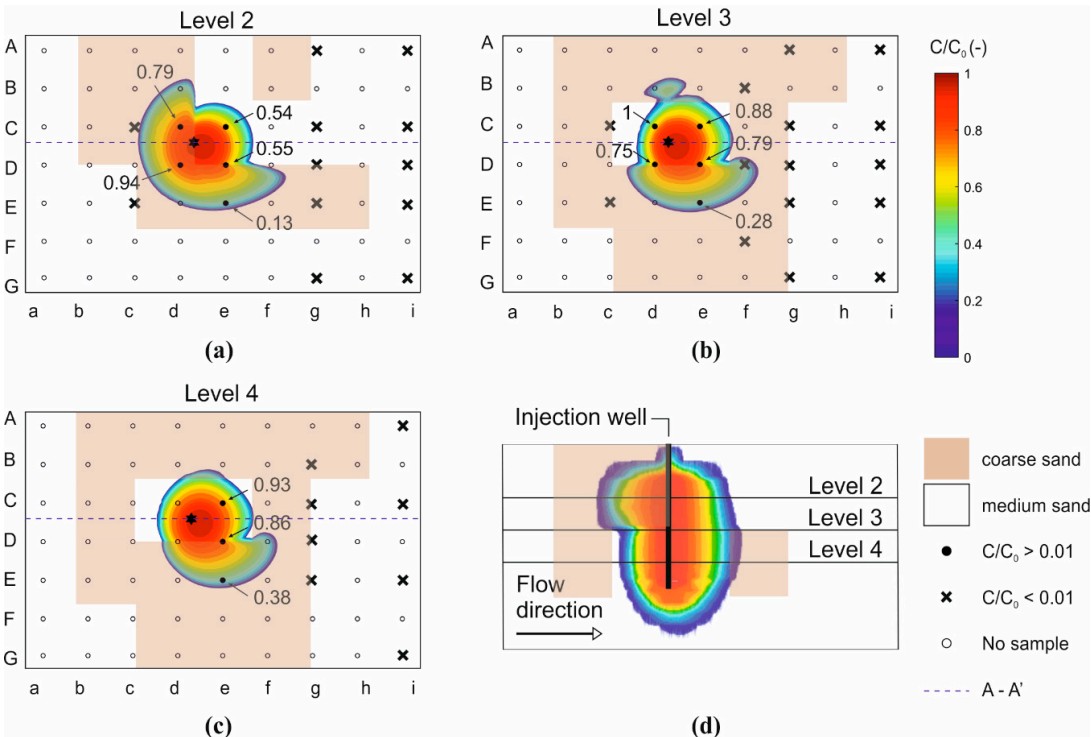

**Figure 3.** Comparison between experimental and modeled concentrations in level 2 (**a**), 3 (**b**) and 4 (**c**) after 8 h from the start of the injection, and modeled spatial distribution of HA-GoeNP spreading area in the transversal section A-A′ (**d**). Modeled concentrations are reported as color maps. Experimental data are reported as numbers where above the detection limit ($C/C_0 = 0.01$), and as a cross where samples were collected but particle concentration was below detection limit. Monitoring points identified as empty circles were not sampled.

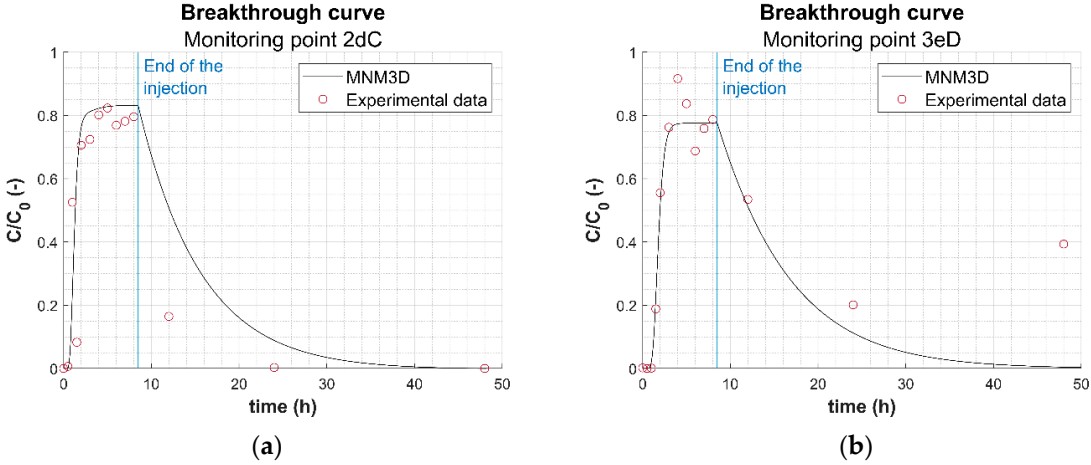

**Figure 4.** Breakthrough curves at monitoring point 2dC ($x_{2dC} = 3.5$ m, $y_{2dC} = 3.84$ m, $z_{2dC} = 3.31$ m) (**a**) and 3eD ($x_{3eD} = 4.5$ m, $y_{3eD} = 2.9$ m, $z_{3eD} = 2.62$ m) (**b**).

*3.2. Change of HA-GoeNPs Size After Injection in A Heterogenous Unconfined Aquifer*

The mean particle size downstream of the injection point significantly increases (Figure 5) at the beginning of the injection and stays for up to 2 to 4 h (3eD and 2eD, respectively). An increase in the mean particle size indicates aggregation and sedimentation of the HA-GoeNPs at the beginning of the

injection that might occur as a consequence of the possible variations in the hydro-geochemistry of the unconfined aquifer. The particle size reduces to almost the initial particle size 4–6 h after the beginning of the injection (e.g., monitoring point 2eD and 3eD), while it is evident that the concentration of particles still increases (e.g., 3eD). This might be caused by the filtration and straining of the HA-GoeNPs in the sand due to the different porosities of the medium and course sand.

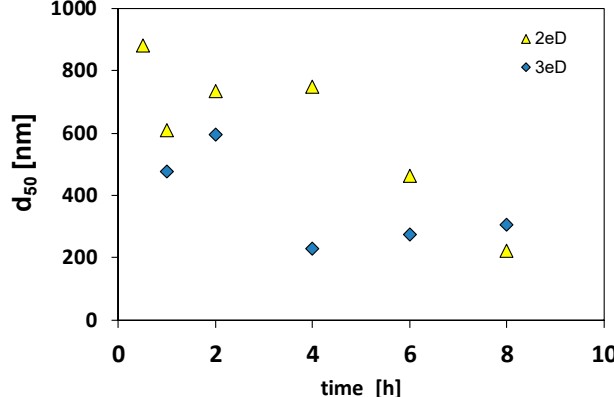

**Figure 5.** Changes in median particle size ($d_{50}$, nm) in water samples downstream the injection point (2eD and 3eD, the coordinates correspond to the ones presented in Figure 1) during HA-GoeNPs injection.

Heterogeneities of the unconfined aquifer greatly affect the transport of the HA-GoeNPs. Hence, before injection, it is beneficial to use modeling tools for assessing the transport of HA-GoeNPs for planning field applications of HA-GoeNPs.

### 3.3. Use of Modeling Tools for Assessing the Transport of HA-GoeNPs

An overview of particle distribution in the container is given by Figure 3, which reports a plan view of simulated and measured concentrations in levels 2, 3 and 4 at the end of the injection (i.e., 8.5 h from the beginning of the simulation). The model results indicate that particles tend to migrate preferentially in the highly permeable areas (i.e., coarse sand), in agreement with experimental data, and that, in this experiment, the permeability distribution affects the particle distribution more than the background flow. This is also clear in Figure 3d, which provides a visualization of the spreading area of suspended particles in the transversal section A-A' crossing the injection well. As evident from the figure, suspended particles efficiently covered the entire depth of the container, with a greater or lesser distribution in accordance with the hydraulic conductivity values of the sand blocks. Modeled and experimental concentrations in Figure 3 show an overall good agreement, which can be investigated in more detail by looking at breakthrough curves in individual monitoring points.

As shown in Figure 4, where breakthrough curves at selected monitoring ports are reported, the model (black line) could correctly reproduce the trend of the experimental data, both in terms of arrival time and shape of the curve. However, in some monitoring points, modelled results slightly overestimated or underestimated the experimental ones (refer also to Figure S8 in Supporting Information for additional breakthrough curves). These discrepancies are more evident for longer times (see Figure 4 and Figure S5); this is confirmed by the coefficients of determination calculated both at individual monitoring points (see Figure S9) and for all points together (Figure S9). The global $R^2$ is higher if calculated for the injection phase only ($R^2 = 0.855$) compared to the entire test ($R^2 = 0.705$); breakthrough curves in individual monitoring points have $R^2$ values systematically higher in the injection phase (Figure S9).

The observed discrepancies between modeled and experimental data are likely due to deviations from the ideal conditions reproduced by the numerical model. Precisely:

- A slight sinking of the particle suspension within the container was observed, especially for long times. Since the model equations do not take into account the gravitational effects induced by the

slurry density and the nanoparticle sedimentation on the long term, the simulated concentration curves may overestimate the actual concentration of suspended particles, particularly in the upper sampling ports. This hypothesis is confirmed also by the less accurate agreement between modeled and experimental breakthrough curves observed in most sampling points after injection (i.e., time higher than 8.5 h), when the experimental concentrations decrease more sharply compared to the modeled curve (see, e.g., Figure 4a and the breakthrough curves reported in Figure S8);

- Changes over time in particle size were observed, as discussed in Section 3.2, which could play a role in particle transport, and are not included in the transport model MNM3D here used to simulate goethite injection;

- The blocks of different sand types in the experimental container may have not been perfectly regular or partial interpenetrations may have existed, and consequently the space-dependent distribution of hydrodynamic properties implemented in the numerical model may be partly inaccurate in some specific areas.

Finally, it is worth noting that the attachment parameter $k_a$, used to simulate particle deposition in the large-scale container, was estimated from column transport tests, and directly applied in the 3D model without further calibration. Consequently, deviations must be expected, even though the overall good agreement between modeled and experimental data suggests that the upscaling approach herein adopted, from column tests to the large 3D container, is able to provide a reliable estimate of particle transport parameters, suitable for at least a preliminary prediction of particle mobility in field-like conditions.

Overall, the simulation of the large container experiment demonstrated that the model was able to successfully replicate the HA-GoeNP transport in the aquifer. This finding is of particular interest since, to the best of our knowledge, the modified advection–dispersion equation for particle transport has never been validated against experiments in three-dimensional large-scale domains. The capability of MNM3D to predict the colloidal particle mobility and transport into the container proved that the code is a valid tool to support the design phase of nanoremediation interventions at field scale applications.

## 4. Summary and Conclusions

In this work, the HA-GoeNP injection and transport within a large-scale container mimicking a heterogeneous aquifer were successfully demonstrated. The particle suspension was stable and mobile during the whole injection, allowing an efficient distribution of the goethite within the artificial aquifer. As expected, the final particle distribution around the well was found heterogeneous, following the pattern of hydraulic conductivity contrast imposed in the container. In particular, longer travel distances were observed inside the higher conductivity zones, where a radius of influence of around 2 m was achieved.

This was the first time that goethite particle injection and transport have been tested in such a realistic setup and controlled environment, demonstrating that HA-GoeNP delivery is feasible for aquifer restoration. The sophisticated monitoring system allowed the gathering of high spatial resolution data of HA-GoeNP concentration during and after the injection. With these data, it was possible to validate, for the first time, the numerical codes MNMs 2018 and MNM3D against reliable 3D particle distributions. In more detail, an experimental-modeling approach was followed to derive the case-specific transport coefficients by least-square fitting the column test results using MNMs 2018. Subsequently, a tracer test was performed in the container and then simulated in MNM3D in order to prove, by comparison between experimental and modeled results, the reliability of the aquifer model in the replication of the real flow field. Finally, the HA-GoeNP transport in the validated model of the aquifer was satisfactorily reproduced using MNM3D. The successful simulation of the large-scale container experiment demonstrated that the numerical solution provided by MNM3D correctly describes the transport of HA-GoeNPs within the container. This finding confirmed that numerical models, and more specifically MNM3D, are valid tools for the simulation of natural and/or engineered micro- and nanoparticle transport in three-dimensional and complex domains. They can be used to support the design phase of nanoremediation interventions at large-scale applications, as they predict the particle behavior in the subsurface, the short and long-term particle spreading, as well as

other important project parameters of the nanoremediation technology. MNM3D can also test possible scenarios of nanoremediation applications, thus reducing the amount of required pilot tests and the consequent global costs of intervention. Given its potential, MNM3D can be applied to bridge the gap among the controlled conditions of laboratory tests and the more complex scenarios typical of field-scale applications.

**Supplementary Materials:** The following are available online at http://www.mdpi.com/2073-4441/12/4/1207/s1, Figure S1: Characterization of the initial HA-GoeNPs. (A) Hydrodynamic radius distributions as estimated with dynamic light scattering (DLS) analysis (triplicate) and (B) Hydrodynamic radius as measured by AF4-MALS and (C) AF4-ICPMS; Figure S2: Breakthrough curves of tracer (tracer test) at the monitoring points (A) 2bA (x2bA = 1.5 m, y2bA = 5.56 m, z2bA = 3.31 m), (B) 4eE (x4eE = 4.5 m, y4eE = 2.12 m, z4eE = 1.84 m). Figure S3: Planar view of the steady state background flow field simulated in Visual Modflow, for the three layers of the large-scale container. Figure S4: Injection of HA-GoeNPs suspension; Figure S5: Spatial distribution of sampling ports used for water samples collection, and comparison between experimental and modeled concentrations; Figure S6: Calibrations to get the NPs concentration from turbidity values. (A) the calibration in deionized water, (B) the calibration in container inflow; Figure S7: Experimental and modeled breakthrough curves for column transport tests. Figure S8: Breakthrough curves of HA-GoeNPs at the monitoring points (A) 2dD (x2dD = 3.5 m, y2dD = 2.98 m, z2dD = 3.31 m), (B) 2eD (x2eD = 4.5 m, y2eD = 2.9 m, z2eD = 3.31 m), (C) 3eE (x3eE = 4.5 m, y3eE = 2.12 m, z3eE = 2.62 m) and (D) 4eE (x4eE = 4.5 m, y4eE = 2.12 m, z4eE = 1.84 m). Figure S9: Correlation graph and global coefficient of determination $R^2$ calculated for modeled and measured particle concentration at all sampling ports where particles were detected; Table S1: AF4 and ICP-MS operational parameters used for HA-GoeNPs characterization.

**Author Contributions:** Conceptualization, C.B., M.V., T.T., and R.S.; methodology, F.v.d.K., C.B., N.F., N.K., and M.V.; software, C.B., N.F., and A.C.; validation, C.B., M.V., T.T., and A.C.; formal analysis, C.B., N.F., and M.V.; investigation, C.B., N.F., K.M., N.K., D.S., S.W., and M.V.; resources, F.v.d.K., T.H., R.S., R.U.M.; data curation, C.B., N.F., and M.V.; writing—original draft preparation, C.B., N.F., and M.V.; writing—review and editing, C.B., M.V., T.T., R.S., R.U.M., A.C., T.H.; visualization, C.B., N.F., A.C., and M.V.; supervision, T.T., R.S., T.H., and F.v.d.K.; project administration, M.V., S.W., C.B., and K.M.; funding acquisition, F.v.d.K., T.H., R.S., and R.U.M. All authors have read and agreed to the published version of the manuscript.

**Funding:** This project received funding from the European Union Seventh Framework Programme (FP7/2007–2013) under Grant Agreement No. 309517.

**Acknowledgments:** Milica Velimirovic is currently a senior postdoctoral researcher of the Research Foundation–Flanders (FWO project number 12ZD120N). The authors gratefully acknowledge the valuable contributions of Sofia Credaro, who assisted in the proofreading and language editing of the manuscript.

**Conflicts of Interest:** The authors declare no conflict of interest. The funders had no role in the design of the study; in the collection, analyses, or interpretation of data; in the writing of the manuscript, or in the decision to publish the results.

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
