# Peer review of "A Large-Scale 3D Study on Transport of Humic Acid-Coated Goethite Nanoparticles for Aquifer Remediation"

_water, doi:10.3390/w12041207_

Round 1

Reviewer 1 Report

This manuscript investigated the use of HA-GoeNPs particles as a remediation agent for field sites and simulated its migration using specialized software: MNM3D & MNMs 2018. The authors performed several transport experiments in columns and in a large scale, heterogeneous 3D aquifer under different flow velocities using either tracer or HA-GoeNPs particles.  They fitted the resulted column breakthrough curves using a developed mathematical model, which is solved with MNM3D & MNMs 2018 software and used the calculated parameters to simulate the transport of HA-GoeNPs particles in the 3D aquifer. The model seems to satisfactorily fit the experimental results. They concluded that the particle delivery around the injection site can make HA-GoeNPs a feasible agent for aquifer restoration and the used software can be used as tool to predict nanoparticle concentration in 3D domains.

The current work is interesting and deserves to be published in Water because it includes sophisticated experiments conducted in field scale realistic 3D aquifer and the results obtained can be used to verify various numerical models. Although there are some issues that need to be addressed and a minor revision is advised.

  • Line 120: please define the abbreviation “bgl.”
  • Lines 172-173: It is stated that the transport eq.1 was solved with the use of MNMs 2018 and MNM3D while there is no mention in the current section how eq. 2 was solved. It has been mentioned only once at the end of Section 2.4.1(lines 218-219) that eq. 2 for the column experiments was solved with MNM3D. It should be clearly stated that both equations (eqs. 1 and 2) for the 3d aquifer were solved simultaneously, as they form a system of differential equations. 
  • Porosity in eq. 1 “ε” used for the 3d, cannot distinguish the difference between coarse and medium sand cases. These two sands have different porosity values in the 3d aquifer. A subscript could be added in porosity symbol. Maybe something like “εΝ”. I assume that the finite difference solver account for the spatial variability of the sand.
  • Lines 218-219: Please rewrite this sentence, especially the last part of it. It is not clear.
  • Lines 232-233: It is stated that the regular grid had dimensions of 900x60x20 cells. Please include a short comment why you chose only 20 cells in the z-direction, while the y-direction has 60 cells. It would be expected to have less cells in the z-direction, if z-direction was much smaller than the y-direction. Is there is an indication of limited migration in that direction?
  • Lines 277-282: The point 3eD indicates that the median particle size rapidly increases with time for the first hour and then reduces, while for the same time, Figure 4b illustrates that the concentration of nanoparticles increases. This is a very interesting finding as it is counter-intuitive, that while the concentration increases the diameter decreases. This observation need more attention. Is there an explanation of how filtration and straining might cause this.
  • Lines 296-303: Here there are given possible explanations of why discrepancies between the experimental and simulated data exist. For the correct model evaluation it should be reiterated here which processes were taken into account by the model and which not. Thusly, while variable conductivity was included in the model, the aggregation and sedimentation processes were not included.

Reviewer 2 Report

This paper is about the use of nanomaterials for in situ remediation of contaminated groundwater. The authors carried out a three-dimensional transport experiment in a large-scale container representing a heterogeneous unconfined aquifer and simulated the experimental conditions by using the numerical code MNM3D (Micro and Nanoparticle transport Model in 33 3D geometries) to predict the particle behavior during the experiment. The HA-Goe nanoparticles were used, revealing a good subsurface mobility, especially within the higher permeability zones.

The paper is well written and experimentally sound. The topic is of great interest and suitable for publication in Water.

The following minor corrections are requested

1) Paragraph 2.4

In lines 129-133 the authors say that injection duration was 8h and that during injection the inlet tank was continuously mixed. It is known that even if the suspension is mixed, the NPs suddenly tend to aggregate loosing partially their properties, so it is impossible to imagine that inlet concentration of active NPs remains constant during injection. On the contrary, since the authors use measures of Fe and turbidity in their experiments, it is acceptable to consider their inlet concentration as a constant.

Anyway the authors should:

  • insert initial and boundary conditions for equations (1) and (2). Is the boundary condition of eqn.1 at the inlet a first or a third type? Is the colloid concentration in the liquid phase at the interface a constant function?
  • Briefly discuss the chosen boundary conditions (also at the exit) in both the 3D and 1D column simulations.

2) Line 206

Is the value of the diameter, equal to 0.25mm, the correct one? If not, please correct it.

3) Use the same units within the paper (mm, cm, m, etc.)

Reviewer 3 Report

The article studies the possibility of using HA-GoeNPs as an effective reagent
in the remediation of contaminated aquifers. To do so, the authors compare results
of transport experiments in a large-scale container representing a heterogeneous
unconfined aquifer and a flow and transport numerical model representing the container to predict the particle behavior.
In particular, the numerical simulations were performed using a numerical code named MNM3D (Micro and Nanoparticle transport Model in 3D geometries).

The article is interesting but some issues that have to be resolved/cleared.
For example, I understand that a single numerical simulation was performed to recreate the transport experiment of Figure 1 and the results are shown in figure 3 and 4.

The questions are:
1. why is it only one simulation to represent the transport experiment? I would have
expected some of them with different values of the parameters, for example, the
hydraulic conductivity representing the different porous medium, etc. and a sensitivity
analysis of the obtained results.

2. line 237: "The background groundwater flow is implementing by 12 fully penetrating wells positioned upstream and by imposing a constant hydraulic head o 3.7 m downstream."
Am I wandering why the wells were situated just at the border of the grid. Did the
Authors check that this fact does not alter the result of the simulations?

3. In the paper is mentioned that
In line 233: where are the 378 monitoring points placed in six sampling planes (1-6)?

4. line 115: "378 sampling points are located in six sampling planes (1-6, numbered from top to bottom), nine sampling rows ... ."
Also, line 140: "HA-GoeNP concentrations were indirectly determined via turbidity measurements performed with ... . The lineare correlation between concentration and turbidity was verified with a five-point calibration in the range from ... ."
I wasn't able to find results of both, head in sampling points and/or turbidity in
a table and/or a figure.

The only figures that represent the results of the numerical simulations are presented in figure 2 and 3. In figure 2 are represented the monitoring points but never presented. I think it would have been interesting to see those results.

Figure 3 shows the comparison between experimental (black isolines) and modeled
(color maps) concentrations in the three levels (a,b,c). They do not seems to fit
too much but without a analysis of the data, I cannot say more than this.
Also in the cross section figure 3 (d), the flow direction is on the right hand side
but coarse sand let the contaminant move to the left. I would have expect the same
behavior on the right hand side on level 4 and below, but does now seems to me that
happens.

Figure 4 represents C/C_0 as a function of the time for only two monitoring points 2dC and 3eD. The Authors shows the comparison between the numerical simulatios results and the experimental data. I do not see pretty much correspondence between them. For example, in figure 4 (a) the red circle does not go until the top and then decades more sharply than the numerical ones. The same happens with figure 4 (b). I do not understand the red circle point in time=48h.

Round 2

Reviewer 3 Report

The authors have satisfactorily responded to all my questions and made the necessary changes to the manuscript.

I think that the paper is ready for publication in Water.